# Effect of Supplementation with Coffee and Cocoa By-Products to Ameliorate Metabolic Syndrome Alterations Induced by High-Fat Diet in Female Mice

**DOI:** 10.3390/foods12142708

**Published:** 2023-07-14

**Authors:** Cheyenne Braojos, Andrea Gila-Díaz, Pilar Rodríguez-Rodríguez, Ignacio Monedero-Cobeta, María Dolores Morales, Santiago Ruvira, David Ramiro-Cortijo, Vanesa Benítez, María A. Martín-Cabrejas, Silvia M. Arribas

**Affiliations:** 1Institute of Food Science Research (CIAL), Universidad Autónoma de Madrid (UAM-CSIC), C/Nicolás Cabrera 9, 28049 Madrid, Spain; 2Department of Agricultural Chemistry and Food Science, Faculty of Science, Universidad Autónoma de Madrid (UAM-CSIC), Ciudad Universitaria de Cantoblanco, 28049 Madrid, Spain; 3Food, Oxidative Stress and Cardiovascular Health (FOSCH) Research Group, Universidad Autónoma de Madrid (UAM-CSIC), Ciudad Universitaria de Cantoblanco, 28049 Madrid, Spain; 4Department of Physiology, Faculty of Medicine, Universidad Autónoma de Madrid (UAM-CSIC), C/Arbobispo Morcillo 2, 28029 Madrid, Spain; 5Confocal Microscopy Unit, Interdepartmental Research Service (SiDI), Faculty of Medicine, Universidad Autónoma de Madrid (UAM-CSIC), C/Arzobispo Morcillo 2, 28029 Madrid, Spain; 6PhD Program in Pharmacology and Physiology, Doctoral School, Universidad Autónoma de Madrid (UAM-CSIC), C/Francisco Tomás y Valiente 2, 28049 Madrid, Spain

**Keywords:** coffee pulp, cocoa shell, polyphenols, caffeine, glycemia, adipocytes

## Abstract

Coffee and cocoa manufacturing produces large amounts of waste. Generated by-products contain bioactive compounds with antioxidant and anti-inflammatory properties, suitable for treating metabolic syndrome (MetS). We aimed to compare the efficacy of aqueous extracts and flours from coffee pulp (CfPulp-E, CfPulp-F) and cocoa shell (CcShell-E, CcShell-F) to ameliorate MetS alterations induced by a high-fat diet (HFD). Bioactive component content was assessed by HPLC/MS. C57BL/6 female mice were fed for 6 weeks with HFD followed by 6 weeks with HFD plus supplementation with one of the ingredients (500 mg/kg/day, 5 days/week), and compared to non-supplemented HFD and Control group fed with regular chow. Body weight, adipocyte size and browning (Mitotracker, confocal microscopy), plasma glycemia (basal, glucose tolerance test–area under the curve, GTT-AUC), lipid profile, and leptin were compared between groups. Cocoa shell ingredients had mainly caffeine, theobromine, protocatechuic acid, and flavan-3-ols. Coffee pulp showed a high content in caffeine, protocatechuic, and chlorogenic acids. Compared to Control mice, HFD group showed alterations in all parameters. Compared to HFD, CcShell-F significantly reduced adipocyte size, increased browning and high-density lipoprotein cholesterol (HDL), and normalized basal glycemia, while CcShell-E only increased HDL. Both coffee pulp ingredients normalized adipocyte size, basal glycemia, and GTT-AUC. Additionally, CfPulp-E improved hyperleptinemia, reduced triglycerides, and slowed weight gain, and CfPulp-F increased HDL. In conclusion, coffee pulp ingredients showed a better efficacy against MetS, likely due to the synergic effect of caffeine, protocatechuic, and chlorogenic acids. Since coffee pulp is already approved as a food ingredient, this by-product could be used in humans to treat obesity-related MetS alterations.

## 1. Introduction

The prevalence of the sedentary lifestyle and inadequate diets has contributed to the development of obesity, which is currently considered a pandemic disease, and the array of alterations which usually cluster with it, hyperlipidemia, hyperglycemia, and hypertension, known as metabolic syndrome (MetS). In turn, MetS is a key factor in the development of cardiovascular diseases, representing one of the most important health problems worldwide [1]. MetS is closely linked to oxidative stress and inflammatory processes, which has prompted the interest in its treatment and prevention based on plant-derived foods containing antioxidant and anti-inflammatory bioactive components with a low toxicity profile and minimal side effects. Among them, polyphenols have been the subject of active research and there is wide experimental and epidemiological evidence showing their capacity to reduce dyslipidemia, fat accumulation, hypertension, and the incidence of MetS [2,3], although there is some controversy regarding hydroxycinnamic acids and weight loss [4].

Food waste is a global problem and strategies to recycle it are gaining relevance. Coffee and cocoa manufacturing are among the food industries generating the largest amounts of by-products and their revalorization is of great interest from the environmental point of view [5,6,7]. Besides other uses, the presence of bioactive compounds in these food matrices suggests their potential benefits in developing supplements and nutraceuticals. Our research group has investigated some of the by-products produced by coffee and chocolate manufacturing, including coffee pulp (the outer part of the coffee cherry), and cocoa shell (produced during cocoa bean roasting). Through milling and green technologies, we have obtained flours and aqueous extracts from coffee pulp (CfPulp-F and CfPulp-E) and cocoa shell (CcShell-F and CcShell-E) containing high levels of phenolic compounds, caffeine, and theobromine, which exhibit antioxidant activity in vitro and in cultured cells [8,9,10]. Ex vivo studies in arteries of aged rats also show that CcShell-E, and some of the abovementioned bioactive compounds, improve endothelial dysfunction through an antioxidant action [11]. In HepG2 cultures, both extracts, as well as the flours derived from these by-products (CfPulp-F and CcShell-F), exhibit hypolipidemic and hypoglycemic properties [12]. We have also demonstrated that these potential new ingredients are safe, and no signs of toxicity were observed in mice after acute (2 g/kg/day, 1 day) or chronic (1 g/kg/day, 90 days) administration [13]. Thus, our own in vitro, cell culture, and ex vivo data suggest that coffee pulp and cocoa shell ingredients have the potential capacity to improve obesity-related MetS alterations. Before they can be incorporated into foods and used to generate products for human consumption, it is necessary to conduct experiments in animal models. Excess fat consumption is one of the most relevant triggering factors leading to MetS, and diet-induced obesity in C57BL/6 mice is a good model mimicking human MetS-related alterations [14,15,16]. Sex-related cardiometabolic risk is still not sufficiently explored and is still controversial. The female sex has usually been regarded as “protected”, but some data indicate that MetS has a higher impact on women in comparison with men. Overall, there is still insufficient information on the effectiveness of therapeutic approaches in women, likely related to the lower number of studies conducted on women and females [17,18].

Our aim was to conduct a comparative study to evaluate the efficacy of supplementation with flours and extracts derived from cocoa shell and coffee pulp to reduce MetS alterations induced by high-fat diet (HFD), to determine which is the ingredient with the best potential for future human applications. In HFD-fed female mice, we have compared the capacity of these four ingredients to reduce body weight, adiposity, hyperglycemia, and hyperlipidemia.

## 2. Materials and Methods

### 2.1. Materials

#### 2.1.1. Equipment

Confocal microscope (Leica TCSSP2, Leica Microsystems, Barcelona, Spain);Glucometer (Accu-Chek Aviva; Roche Diagnostics, Penzberg, Germany);Hewlett-Packard-1100 HPLC-diode array detector (DAD) chromatograph (Agilent Technologies, Palo Alto, CA, USA); a mass spectrometer (MS) was coupled to the HPLC system, and the detection was conducted in an API-3200 Qtrap (Applied Biosystems, Darmstadt, Germany).

#### 2.1.2. Materials and Reagents

Acetonitrile (Panreac Química S.L.U. (Barcelona, Spain);CITIFLUOR-AF (Citifluor Ltd., London, UK);Cocoa shell, supplied by Chocolates Santocildes Ltd. (León, Spain);Coffee pulp, supplied by Supracafé Ltd. (Madrid, Spain);DAPI (Invitrogen, ThermoFisher, Madrid, Spain) to stain nuclei;Diet-Control (Euro Rodent Diet 22; 5LF5, Labdiet, Madrid, Spain);Diet-HFD (Altromin C 1090; Altromin, Lage, Germany);Flavoring agent (vanilla flavor; MyProtein, Hut.com Ltd., Manchester, UK);Formic acid (Panreac Química S.L.U., Barcelona, Spain);Gelatin (Inkafoods, S.L., Barcelona, Spain);D-(+)-Glucose (Merck Life Sciences S.L.U, Madrid, Spain);Heparin (Rovi, Madrid, Spain);HPLC columns (Spherisorb S3 ODS-2 C8 column; Waters, Milford, MA, USA);Lipid profile (Kit, Spinreact, Barcelona, Spain);Leptin ELISA (Kit; BioVendor R&D R, Brno, Czech Republic);Mitotracker red FM (Invitrogen, ThermoFisher, Madrid, Spain);Mice (adult C57BL/6 J females) bred at the Animal House Facility of the Universidad Autónoma de Madrid (Madrid, Spain);Paraformaldehyde (Panreac Química S.L.U., Barcelona, Spain);Porcine pancreatin (Sigma-Aldrich, Madrid, Spain);Porcine pepsin (Sigma-Aldrich, Madrid, Spain).

### 2.2. Methods

#### 2.2.1. Preparation of Food Ingredients from Coffee and Cocoa By-Products

Coffee pulp and cocoa shell were ground, yielding the respective flours (CfPulp-F and CcShell-F). The flour was maintained in sealed flasks at −20 °C until use. The coffee pulp and cocoa shell aqueous extracts (CfPulp-E and CcShell-E) were prepared according to sustainable green aqueous extraction conditions previously optimized [8,9]. Briefly, the CfPulp-E and CcShell-E (0.02 g/mL solid-to-solvent ratio) were added to boiling water (100 °C) and mixed for 90 min. Thereafter, the aqueous extracts were filtered, frozen at −20 °C, freeze-dried, and stored at −20 °C until further usage.

The four ingredients were subjected to an in vitro simulated static digestion following the INFOGEST procedure with minor adjustments [19]. In brief, the abovementioned extracts (0.1 g) and flours (1 g) were combined with a simulated oral fluid and stirred (2 min, 37 °C) to mimic the oral fraction. The resulting oral phase was mixed with the simulated gastric fluid and pepsin (2000 U/mL) and incubated under continuous stirring (2 h, 37 °C). After that, the simulated gastrointestinal fluid including pancreatin (100 U trypsin/mL) was added and incubated (2 h, 37 °C). All digestions were performed in duplicate. The bio-accessible fractions (supernatants) collected from each digestion phase were freeze-dried and stored at −20 °C until utilization.

#### 2.2.2. Characterization of Food Ingredients from Coffee and Cocoa By-Products

Phenolic compounds and methylxanthines of the four ingredients were analyzed using a Hewlett-Packard-1100 HPLC system with a diode array detector (DAD). As mobile phases, 0.1% formic acid in water (solvent A) and 100% acetonitrile (solvent B) were utilized using the following gradient: 15% B for 5 min, 15–20% B for 5 min, 20–25% B for 10 min, 25–35% B for 10 min, 35–50% B for 10 min, and column re-equilibration. Injection volumes of the samples were between 10–20 mL. The chromatographic separation of phytochemicals was conducted at a flow rate of 0.5 mL/min at 35 °C in a Spherisorb S3 ODS-2 C8 column (Waters, Milford, MA, USA) (3 μm, 150 mm × 4.6 mm i.d.). The preferred wavelengths for DAD detection were 280 nm (hydroxybenzoic, hydroxycinnamic acids, and caffeine) and 370 nm (flavonols and flavones). A mass spectrometer (MS) was coupled to the HPLC system, and the detection was conducted in an API-3200 Qtrap (Applied Biosystems, Darmstadt, Germany) equipped with an ESI source, triple quadrupole-ion trap mass analyzer, and Analyst 5.1 software.

The phenolic compounds and methylxanthines were characterized by their retention times, UV and mass spectra, and fragmentation patterns compared to authentic standards when available. For quantitative analysis, amino derivatives of caffeic and *p*-coumaric acid were quantified by calibration curves of the corresponding free acid. Apigenin 6-C-glucoside was used for C-glycosides flavones derived from apigenin, with flavonols derivatives of quercetin by the curve of quercetin-3-O-glucoside. Theobromine and caffeine were quantified by the calibration curves of their respective standards.

#### 2.2.3. Experimental Design in Mice

The experiments were approved by the Ethics Review Boards of Universidad Autónoma de Madrid (CEI-UAM 96-1776-A286) and the Regional Environment Committee of the Comunidad Autónoma de Madrid (Madrid, Spain; PROEX 108-19, date of approval 16 May 2019).

Thirty C57BL/6 J mice were used. The inclusion criteria were sex (female), a similar body weight (average weight 16–18 g), and age (seven-week-old) at the beginning of the study. The mice were divided into 6 groups (5 mice/group) and housed under controlled light (12 h light/dark cycles from 8 am to 8 pm) and temperature (22–24 °C) conditions. 

For 6 weeks, the Control group was fed with standard diet containing 4.4% fat, 55% carbohydrates, 22% protein, and 4.1% fiber, and the remaining 5 groups were fed with an HFD containing 60% fat, 24% carbohydrates, 16% protein, and 3.8% fiber. Drinking water was provided ad libitum in all groups. Thereafter, in HFD-fed mice, supplementation protocol was performed for another 6 weeks with one of the ingredients (included in a gelatin matrix; see preparation below) forming the following groups: HFD-CfPulp-E, HFD-CfPulp-F, HFD-CcShell-E, and HFD-CcShell-F. Another HFD-fed group and the Control group were supplemented with a neutral gelatin during the same period. The mice were weighed weekly throughout the study, and growth slope of each mouse was calculated. Fasting glycemia was assessed at 3 time points, at the beginning of the study, after 6 weeks with HFD alone, and at the end of the supplementation protocol (week 12). Glucose tolerance test (GTT) was performed at week 12, at the end of the study (one day prior to sacrifice).

At the end of the supplementation period, the mice were killed by standard procedures approved by the Animal Ethical Committee. They were first included in a chamber with carbon dioxide to induce hypoxia and avoid pain, followed by exsanguination by cardiac puncture. Blood was collected in tubes containing 5% heparin. Finally, a sample of mesenteric white adipose tissue (WAT) was collected and stored in 4% paraformaldehyde.

A graphical summary of the experimental design protocol is shown in Figure 1.

### 2.3. Supplement Preparation and Supplementation Protocol

After 6 weeks with standard diet or HFD, supplementation was initiated using gelatin cubes and voluntary ingestion, as we have previously described in rats [20]. During this period, mice were maintained on HFD (except the Control group which was kept on standard diet).

Gelatin cubes were prepared with 100% bovine gelatin in water at a concentration of 140 g/L. Once the gelatin was dissolved in hot water, vanilla flavor (4.8 mL/L) and one of the extracts or flours were incorporated at the appropriate dose, or the gelatin was maintained neutral without ingredient. The mixture was transferred to a mold to prepare 1 cm^3^ size cubes and left to cool down to allow solidification. The individual cubes were extracted and frozen in plastic bags until they were used. 

Supplementation was performed at a dose of 500 mg/kg/day (5 days a week, for 6 weeks). To adjust the dose, each mouse was weighed weekly, and the cubes were prepared according to the weight for the week. The procedure consisted of placing the mouse individually in an empty box without bedding with the cube and left until it was fully ingested, usually in less than 5 min. Thereafter, the mouse was returned to the box.

### 2.4. Plasma Lipid Profile and Leptin Assays

Blood was centrifuged for 10 min (900× *g* at 4 °C) and the plasma was aliquoted and stored at −70 °C to evaluate lipid profile and leptin. Assessment of total, low-density lipoprotein cholesterol (LDL) and high-density lipoprotein cholesterol (HDL), and the quantification of endogenous leptin were determined using specific kits following the manufacturer’s protocols.

### 2.5. Glucose Tolerance Test (GTT)

The mice were fasted for 6 h before the glucose load (i.p. bolus of 1 g/kg at time 0). Blood glucose level was measured before (basal) and at 15, 30, 45, 60, and 90 min after injection. Glucose determination was performed in blood samples obtained from the tail with an Accu-Chek Aviva glucometer. Area under the curve (AUC) was used to assess differences between groups.

### 2.6. Adipocyte Study

Perivascular adipocytes were dissected from the mesenteric bed and studied with confocal microscopy as previously described [21]. Briefly, to evaluate adipocyte size, the WAT samples were mounted on a slide with a small well filled with CITIFLUOR-AF mounting medium and were visualized with a spectral laser scanning confocal microscope. Single images from 5 randomly chosen regions were captured with a ×40 objective at Ex. 488 nm/Em. 500–560 nm to visualize adipocyte size based on their autofluorescence properties. Quantification was performed by FIJI free software (ImageJ version 1.53c, Wayne Rasband, Java 1.8.0/172 NIH; USA) [22]. To analyze adipocyte size, the number and area occupied by the adipocytes were measured in the images and the average adipocyte size was calculated from them.

We evaluated browning through mitochondria staining. For this, adipocytes were incubated with Mitotracker red 1:2000 in phosphate buffer saline (1:4000, 60 min, darkness, room temperature), followed by staining with 4′,6-diamidino-2-phenylindole (DAPI) to stain nuclei (1:500 *v*/*v* from stock solution, 15 min, darkness, room temperature). After 2 washing periods of 20 min, the adipocytes were mounted as indicated above and visualized at Ex. 405 nm/Em. 410-475 (DAPI staining nuclei), Ex. 488 nm/Em. 500–560 (autofluorescence), and at Ex. 581 nm/Em. 644 nm (Mitotracker) under the same laser %, brightness, and contrast conditions. In Mitotracker images, the area and intensity of adipocytes were measured, and, from them, the integrated density was calculated.

### 2.7. Statistical Analysis

Statistical analysis was performed by R software (version 3.6.0, 2018, R Core Team, Vienna; Austria) within the RStudio interface using *rio*, *dplyr*, *compareGroups*, *ggpubr*, *devtools*, and *ggplot2* packages. Significance was established at a *p*-value < 0.05.

The bioactive compounds in the tested ingredients were described as the mean ± standard deviation using, at least, three experiments. The statistical analysis of the comparison between raw and digested fractions was performed with one-way analysis of variance (ANOVA), followed by post hoc Tukey’s test, to compare the gastrointestinal fraction. Mice data were expressed as the median and interquartile range [Q1; Q3] according to Kolmogorov–Smirnov test, and Kruskal–Wallis by ranks and Hold post hoc were used to test the significance in the independent experimental groups. Spearman coefficient (rho) between leptin and adipocyte size were also calculated.

## 3. Results

### 3.1. Bioactive Components of the Ingredients

The bioactive composition of coffee pulp and cocoa shell extracts and flours were analyzed in the raw and the digested fractions (Table 1). The comparison between the four ingredients showed that the phytochemical composition was different between coffee pulp and cocoa shell, as well as between extracts and flours. 

Methylxanthines were the major phytochemicals in both coffee pulp and cocoa shell matrices. In both coffee pulp ingredients, caffeine was the only methylxanthine found, its content being 1.7-fold higher in the extract than in the flour. Although caffeine content was reduced by gastrointestinal digestion, 85% persisted in CfPulp-E and 82% in CfPulp-F. In cocoa shell matrices, theobromine was the main methylxanthine, its content being 4.9-fold higher in the extract with respect to the flour, whereas caffeine content was higher in the flour (5-fold). After in vitro digestion, both theobromine and caffeine levels were reduced to 50% in the CcShell-E, while the levels were better maintained in the CcShell-F (theobromine 97% and caffeine 78%).

Regarding phenolic compounds, CfPulp-E displayed the highest total levels, followed by CfPulp-F, and the cocoa shell ingredients had the lowest levels. Interestingly, both by-products shared a high content in hydroxybenzoic acids, with the major protocatechuic acid in the coffee pulp, and gallic acid in the cocoa shell ingredients. With respect to hydroxycinnamic acids, the pattern was different; chlorogenic acids were the principal phenolic compounds found in coffee pulp ingredients, always showing higher levels for CfPulp-E than CfPulp-F. N-phenylpropenoyl-L-amino acids were only detected in cocoa shell samples, and CcShell-E exhibited 3.7-fold higher levels compared to CcShell-F. With respect to flavonoids, vicenin-2 was present in all samples, except for CcShell-F, showing the highest levels in CfPulp-E. As for flavan-3-ols (catechin and epicatechin), they were only present in cocoa shell ingredients, representing 25% of total phenolic compounds in CcShell-E, and 28% in CcShell-F. Catechin was the main flavanol detected (93% in CcShell-E and 89% in CcShell-F), followed by epicatechin. Finally, flavonols were present in all samples but in low contents (1.4–4.7% of total phenolics); rutin and isoquercetin were detected in coffee pulp, and quercetin glucosides in cocoa shell ingredients.

In addition, our results revealed the transformation or degradation of phenolic compounds which takes place through the course of the simulated digestion with different degrees of persistence. Regarding hydroxybenzoic acids, gallic acid seemed to be more susceptible to be degraded than protocatechuic acid in all coffee pulp and cocoa shell ingredients. The persistence of protocatechuic acid ranged from 71 to 96% with respect to raw samples. Some hydroxycinnamic acids were not detectable after simulated digestion, namely, 5-feruloylquinic and 3,5 dicaffeoylquinic. Others were still present, but significantly reduced, like cryptochlorogenic acid (*trans*) in coffee pulp and N-caffeoyl-L-aspartate in cocoa shell, and the rest were only slightly and not significantly reduced by digestion. Regarding the impact of digestion on flavan-3-ols found in cocoa shell ingredients, catechin decreased in the extract, while, in the flour, it was significantly increased after the digestion process, and epicatechin was maintained in the extract while it was lost in the flour. Flavonols (rutin, isoquercetin, and quercetin derivatives) only remained in CfPulp-F with significantly lower levels and were not detected in the rest of the ingredients after digestion.

### 3.2. Effect of Supplementation on Body Weight and Growth Rate in HFD-Fed Female Mice

The initial body weight was similar in all the groups, with no statistical difference between them. At week 6, body weight in Control mice was not significantly different compared to initial weight. However, all the HFD groups had increased body weight, being significantly larger compared to Control, with no significant differences among the five HFD groups, prior to supplementation (Table 2). 

At week 12, after supplementation, all HFD groups had significantly larger body weight compared to the Control group, without significant differences between the different supplementation protocols (Figure 2A). Body growth rate, calculated as the slope of the lineal equation of each mouse, was significantly higher in all HFD groups compared to Control. Compared to HFD, the slope was lower in mice supplemented with coffee ingredients, being significant only in the CfPulp-E group (Figure 2B).

### 3.3. Effect of Supplementation on Adipocytes and Plasma Leptin in HFD-Fed Female Mice

At week 12, the end of the supplementation period, WAT adipocytes were significantly larger in the HFD group with respect to Control. Supplementation with both coffee pulp ingredients normalized adipocyte size to levels non-significantly different from the Control group (Figure 3). Supplementation with CcShell-F significantly reduced adipocyte size compared to HFD, but adipocytes were still larger compared to the Control group.

Mitotracker staining was significantly reduced in HFD compared to the Control group. Only the CcShell-F group had a significantly larger staining compared to HFD (Figure 4). This was not possible to study in CfPulp-E due to insufficient tissue in some mice.

At the end of the supplementation period, leptin levels were significantly higher in all HFD groups compared to Control. Leptin levels tended to be lower in HFD + CfPulp-E compared to HFD, near statistical significance (*p*-value = 0.059), and were not modified by the other by-products (Table 3). A significant and positive correlation was found between adipocyte size and leptin levels (rho = 0.70; *p*-value < 0.001).

### 3.4. Effect of Supplementation on Glucose Homeostasis in HFD-Fed Female Mice

At the start of the study, basal fasting glycemia was not significantly different among groups and, by week 6, all HFD groups had a significant elevation compared to the Control group (Table 4).

At week 12, the end of the supplementation period, HFD mice maintained a significantly larger basal glycemia compared to the Control group. Supplementation with CcShell-F, CfPulp-E, and CfPulp-F significantly reduced basal glycemia compared to HFD, to levels not significantly different from the Control group (Figure 5A). The GTT AUC was also significantly higher in HFD with respect to the Control group. Cocoa shell ingredients (extract or flour) did not significantly modify AUC with respect to HFD. However, both coffee ingredients significantly decreased it (Figure 5B). It was not possible to obtain sufficient plasma to evaluate insulin.

### 3.5. Effect of Supplementation on Lasma Lipid Profile at in HFD-Fed Female Mice

At week 12, the end of supplementation, total cholesterol was significantly higher in HFD-fed mice compared to the Control group. No significant difference in any of the supplemented groups was found compared to HFD (Figure 6A). LDL was also significantly increased in HFD with respect to the Control group; no differences were observed in the supplemented groups compared to HFD or Control (Figure 6B). HDL was not different between the HFD and Control groups; both cocoa shell ingredients (extract and flour) and CfPulp-F increased HDL levels compared to Control and to the HFD groups (Figure 6C). Triglyceride levels were not different between Control and HFD. Both coffee pulp ingredients tended to reduce triglyceride levels compared to Control, being significant only in CfPulp-E (Figure 6D).

## 4. Discussion

Coffee and cocoa manufacturing are among the largest food industries, also producing massive amounts of waste. According to the International Coffee Organization, about 171.3 million 60 Kg bags of coffee beans are estimated to be produced in 2023, and global cocoa production is around 5 million tons annually [23,24]. In the forthcoming years, an increasing trend is expected due to the rising of consumer’ demands, and, consequently, their waste products will also go up. We have previously demonstrated that some of the by-products from these industries—namely, cocoa shell and coffee pulp—contain bioactive compounds with antioxidant and ant-inflammatory properties, and exhibit hypolipidemic and hypoglycemic properties in cell cultures. Therefore, our aim was to compare the efficacy of aqueous extracts and flours from these by-products to reduce metabolic alterations associated with obesity induced by HFD in C57BL/6 female mice. We chose this intervention since inadequate diets represent the main cause of obesity and associated cardiometabolic alterations, which cluster in the MetS. We used HFD-fed mice as the experimental model since they develop alterations resembling MetS in humans, including increased fat, hyperglycemia, and an altered lipid profile [14,15,16]. We focused on females, since the female sex is under-represented in animal studies as well as in clinical trials, as it is important to gain knowledge about the impact of HFD and the capacity of food ingredients to counteract MetS alterations in women [17].

### 4.1. Comparison of Bioactive Compounds Found in the Ingredients and the Effect of Digestion

In previous studies, we have optimized extraction methods of the by-products used in the present study and fully characterized their content in bioactive molecules, mainly methylxanthines and polyphenols [9,10]. In the present study, we also analyzed the composition of cocoa shell and coffee pulp flours and extracts, and conducted a comparative analysis, also evaluating the influence of gastrointestinal digestion, which may better reflect the final bioactive molecules reaching plasma and exerting effects in vivo. In cocoa shell ingredients, the main methylxanthine found was theobromine, with its content in the aqueous extract being especially relevant, with levels like those found in other alcoholic and aqueous extracts [25]. Both matrices also contained significant amounts of caffeine, which was higher in the flour. CcShell-E contained higher total phenolic content than the flour, with hydroxybenzoic acids (gallic and protocatechuic acids) being the most abundant, followed by flavan-3-ols (catechin and epicatechin), which are characteristics of this by-product not present in coffee pulp, in accordance with our previously published data on the optimization and characterization of CcShell-E bioactive components [9]. Other studies analyzing cocoa shell with HPLC-MS have evidenced that procyanidins and catechins are the main polyphenols in this by-product with a total phenolic content ranging from 6.04 to 94.9, mg GAE/g, but these authors did not detect hydroxybenzoic acids, while they were found in cocoa husk, another cocoa by-product [26]. Compared to other studies, our data showed a higher total content of phenolic compounds in CcShell-E, likely due to differences in the method of extraction; we used heat-assisted extraction, which may have facilitated the solubilization of the phytochemicals from the ingredients [27], and used previously optimized conditions [9].

In coffee pulp ingredients, we found caffeine to be the main methylxanthine. Both caffeine and total phenolic content were higher in this by-product compared to cocoa shell, and the variety was larger, including hydroxybenzoic and hydroxycinnamic acids, flavones, and flavonols. Our results are in accordance with previous studies, including ours, analyzing the composition of this by-product which indicate that the main bioactive components were caffeine and phenolic compounds, highlighting the presence of chlorogenic and protocatechuic acids [10,28]. In comparison with coffee, in coffee pulp, we found chlorogenic and ferulic acids, but not caffeic or *p*-coumaric also present in coffee [29]. Coffee is usually consumed after roasting, a process which reduces the degree of bioactive compounds, that is, caffeine, and chlorogenic acid decreased even to half [29]. For example, total phenolic content in light roasted coffee ranges from 59.8 to 38.3 g GAE/kg and in dark roast from 37.4 to 47.4 g GAE/kg [30]. Our data with coffee pulp extract show a content of 597.6 mg GAE/100 g, which is comparatively higher.

We also evaluated the modification in the content of bioactive components after simulated digestion. Overall, the digested ingredients maintained a high profile of bioactive compounds, although they were mostly reduced. After digestion, both theobromine and caffeine contents were markedly reduced in the extracts, while they decreased to a lesser extent in flours, due to the presence of the fiber matrix that protects these compounds from degradation throughout the digestive process [31]. Similar behavior was observed for phenolic compounds. It is well-known that phenolic compounds are strongly linked to cell walls, mainly polysaccharides, compromising their bioaccessibility as well as protecting them throughout digestion [32]. Our results revealed that, under the enzymatic conditions of the in vitro simulated digestion, the phytochemicals from the flours seemed to release slowly and continuously, which could maintain a higher phenolic concentration. For example, catechin increased in the CcShell-F after digestion. 

### 4.2. Efficacy of the Ingredients on Body Weight and Adipose Tissue

In adult female mice, HFD induced an increase in body weight, as previously shown in males [33,34]. The growth increase was larger in the first 6 weeks of HFD and, thereafter, by week 12, the body weight increase slowed down, indicating that the animal was reaching a steady state of growth. None of the ingredients reduced weight, although the slope of growth was lower with coffee pulp extract. It is possible that a higher dose of CfPulp-E may have achieved a significant weight reduction, as described in male rats fed with HFD under similar experimental conditions, supplemented with a dose of 1000 mg/kg/day [35]. A longer duration of supplementation may also induce larger effects. This is supported by data on randomized clinical trials showing that only long-term coffee consumption improves body mass index, body fat, and other MetS alterations in adult women and men [36,37], adolescents [38], and perimenopausal women [39].

Even though body weight was not improved by supplementation, CcShell-F, CfPulp-E, and CfPulp-F were able to reduce adipocyte size, with coffee pulp ingredients being more effective, normalizing size. Regarding the bioactive components responsible for the reduction of adipose tissue, we suggest that caffeine is a good candidate, since the abovementioned ingredients showed a high content. There is consensus that caffeine intake is effective in fat reduction, evidenced by a systematic review and meta-analysis of randomized controlled trials [40]. A recent randomized clinical trial comparing two weight-loss interventions with coffee (160 mg caffeine) or green tea (252 mg catechin and 96 mg caffeine) revealed a significant advantage with coffee consumption, suggesting a potential benefit of caffeine [38]. This is confirmed by other clinical trials showing that the consumption of a caffeinated formula was more effective in reducing the percentage of fat compared to a decaffeinated one [41]. The fat-reducing effect of caffeine can be explained through several mechanisms: the release of noradrenaline and dopamine, stimulating neuronal activity in several brain regions, the increase fat oxidation through the inhibition of phosphodiesterase, and a thermogenic effect [40]. In addition to caffeine, other bioactive components may account for the reduction in adipocyte size since we observed a larger effect with coffee pulp ingredients. We suggest that chlorogenic acid and related compounds may contribute to the effect since CfPulp-E had the highest content and these hydroxycinnamic acids were absent in cocoa shell ingredients. In addition, in our previous study, in cultured adipocytes with ingredients from other coffee by-products (Husk and Silverskin), chlorogenic acid was responsible for reduced adipogenesis and lipid accumulation [42], and, in male animal models of obesity and hypercholesterolemia, chlorogenic acid was effective in reducing fat accumulation in the liver [43,44]. It must be noted that there is some controversy regarding the effect of hydroxycinnamic acids on body weight control. In a very large European cohort of nearly 400,000 individuals evaluating the role of dietary polyphenols on body weight change over a 5-year period, most of the polyphenols showed a negative association, but a positive trend or no effect was found with 3-feruloylquinic, 4-caffeoylquinic, and 4-feruloylquinic acids, with coffee being the main dietary source [4]. Although derivatives from these hydroxycinnamic acids were found in raw coffee pulp matrices, these components were lost after in vitro digestion, suggesting that they are not likely to play a major role. Protocatechuic acid is another bioactive component present in the tested ingredients which may have contributed to the reduction in adipocyte size, since male rats fed with a similar HFD protocol for 14 days, supplemented with protocatechuic acid at 50 mg/kg/day, was able to reduce adipocyte size enlargement [45].

In addition to an effect of the ingredients through a reduction in adipogenesis and lipid accumulation in adipocytes, stimulation of the browning process may also participate in adipocyte size reduction, since WAT is interspersed with thermogenic brown adipocytes, which have a larger energy expenditure [46]. We have previous evidence that caffeine can induce adipocyte browning in cell cultures [42], increasing UCP1 protein [47]. Theobromine [48] and protocatechuic acid can also stimulate brown adipose tissue [49]. In our study, we evaluated browning through mitochondria staining, which are abundant in brown adipocytes [46]. We found that only cocoa shell flour was effective, probably related to the combination of caffeine, theobromine, and protocatechuic acid. We suggest that the increase in energy expenditure through the browning process can contribute to the observed adipocyte size reduction induced by cocoa shell flour.

A third possible mechanism contributing to the observed reduction in adipocyte size by supplementation could be the induction of satiety, as it has been proposed that non-caffeine ingredients in coffee may have the potential to decrease body weight, through this mechanism [50]. We could not test the amount of food ingested daily by the animals, since HFD mice left a large amount of food debris inside the cage, making it impossible to accurately quantify food intake.

Together with adipocyte enlargement by HFD, we observed a concomitant hyperleptinemia, also characteristic of obesity, which we have previously evidenced in male mice with the same obesogenic diet [33]. The largest reduction of adipose tissue was observed with coffee pulp extract together with a concomitant reduction in leptin. Despite the normalization of adipocyte size by this ingredient, plasma leptin levels did not reach control levels, suggesting it needs a longer period to normalize hormonal levels after fat loss.

### 4.3. Efficacy of the Ingredients on Glucose Homeostasis

HFD induced hyperglycemia and alterations in GTT in females, as we have previously described in males [16,33], suggesting similarities between the sexes in the response to HFD. Our previous studies also showed hyperinsulinemia in this mouse model, which was not possible to evaluate in the present study due to insufficient plasma sample in some mice. We demonstrated that all the ingredients, except cocoa shell extract, were equally effective in the restoration of basal glycemia to control levels, and even this ingredient improved this parameter, although not significantly. Moreover, both coffee pulp ingredients were able to normalize GTT-AUC, suggesting their potential as antidiabetic ingredients, also demonstrated in obese male rats [51]. In humans, regular coffee consumption has been associated with an improvement of glucose metabolism, reducing the risk of diabetes [36,52]. However, the role of caffeine intake in this association has remained unclear and epidemiological studies and randomized clinical trials using caffeinated and decaffeinated coffee drinks show that the protective effect of coffee may not be directly related to caffeine, and other coffee components may be relevant [41,52]. Chlorogenic acid, present in coffee and in the coffee pulp ingredients tested in the present study, may participate in this response. This is suggested by the glucose-lowering effects of coffee silverskin, another coffee by-product with a similar composition to coffee pulp [53], and the demonstrated antidiabetic properties of chlorogenic acid, through the reduction of gastrointestinal absorption of glucose and activation of AMPK-induced glucose transporters [54,55,56]. Another component present in the tested ingredients with hypoglycemic properties is protocatechuic acid, which has been shown to improve systemic insulin resistance in HFD-fed male mice [57].

### 4.4. Efficacy of the Ingredients on Lipid Profile

HFD increased total and LDL cholesterol, as previously described in other studies in mice [15,33,34]. None of the ingredients were able to reduce these parameters, although we observed that CfPulp-E reduced triglyceride levels, like the findings in diabetic rats [35]. It is possible that the dose used in the present study was not sufficient, since coffee pulp at 1000 mg/kg/day improved the lipid profile in hypercholesterolemic rats [35]. Regarding the bioactive compounds implicated, Ontawong et al. suggested the beneficial effects of coffee pulp were related to the content of chlorogenic acid [35], which has lipid-lowering effects [44], through a reduction in cholesterol micelle transport [35], inhibition of pancreatic lipase [58], or inducing lipolysis [59]. In rats, treatment with coffee silverskin (containing caffeine and chlorogenic acid) was also able to reduce cholesterol and triglycerides [53]. The differences between these studies in rats and ours in mice may be related to variances between these experimental models regarding plasma lipid metabolism. Another possible explanation is differences in other bioactive components with effects on cholesterol, such as ferulic acid (a major metabolite of feruloylquinic acids), which reduced total cholesterol, LDL cholesterol, and triglycerides in diabetic rats [60]. It must be noted that, in our extracts and flours, this component was lost after digestion. Even though our experimental model did not show a reduction in HDL, we found that cocoa shell ingredients increased it, which could be a beneficial action of this by-product. An increase in HDL with protocatechuic acid supplementation has also been demonstrated in male rats exposed to HFD [61] and, in male hamsters receiving a high-cholesterol diet, this compound also evidenced cholesterol-lowering effects [62].

### 4.5. Influence of Sex

We studied the impact of the potential by-product in female mice, which are under-represented in studies regarding HFD-induced alterations. We demonstrated that coffee pulp, particularly the extract, was the best ingredient, and was able to improve, or even normalize many of the MetS alterations induced by HFD. An important question is whether our results can be extrapolated to male mice and if this ingredient would benefit to the same extent men and women. The present findings in HFD-fed female mice are quite similar to those found in male mice with the same protocol [16,33]. Other authors using male and female C57BL/6 mice fed with HFD (60% fat) + sucrose over 12 weeks also showed a similar body weight increase, fat mass content, and lipid profile suggesting similarities in the response to HFD between sexes [63]. Based on the similar responses to HFD of male and female mice, we can predict that the beneficial effects of coffee pulp ingredients on MetS alterations in female mice could be extrapolated to male counterparts. This is also supported by the fact that in males exposed to HFD, coffee pulp supplementation has also been able to reduce body weight gain, and result in an improved lipid profile [35] and glucose homeostasis [51]. Moreover, supplementation with specific components present in coffee pulp has also shown positive effects in male experimental models of MetS. For example, chlorogenic acid reduced fat accumulation in male rats fed with a high-cholesterol diet [44], and protocatechuic acid improved insulin resistance and lipid profile in males from various species [57,61,62]. Despite this evidence, it would be interesting to evaluate the impact of the tested by-products in male HFD-fed mice to confirm if similar responses occur.

The positive results with coffee pulp observed in the present study suggest that this by-product could be also used in humans to ameliorate MetS, raising the question of whether women and men could equally benefit from this supplementation, since there are important differences in the development of MetS between sexes, particularly related to sex hormone differences and the influence of menopause. MetS is slightly higher in adult men, but markedly increases in women after menopause due to the protective effects of estrogens against MetS [64]. This is also confirmed in ovariectomized female mice, which show an exacerbated HFD-induced obesity [65]. Moreover, differences in the risk factors for cardiometabolic disease development differ between sexes. In men, hypertension seems to be the predominant risk factor while, in women, the main factors are low HDL and high triglycerides levels [66]. Fasting plasma glucose is a higher risk factor for all-cause mortality in women compared men [67]. Our results in female mice showing fat mass reduction and improving lipid profile after coffee pulp supplementation suggest that women, particularly after menopause, may benefit more than men from the use of this by-product. This is also supported by evidence that habitual coffee consumption protected from MetS in both sexes, but women showed better results [37]. This needs to be confirmed by clinical trials. The recent acceptance of this by-product as a novel food and safe ingredient for preparing non-alcoholic drinks and infusions pursuant to E.U. regulation 2015/2283 [68] suggests that it can be safely used in humans.

In summary, among the tested food ingredients, coffee pulp, particularly the extract, had the greatest efficacy in improving the alterations induced by HFD, reducing the rate of weight gain and leptin and normalizing adipocyte size and glucose alterations. Although coffee pulp did not influence cholesterol levels, an increase in HDL and a reduction in triglycerides was also observed. Cocoa shell flour, which also contained a higher content of bioactive components also showed improvement in some of the parameters. We summarize our main findings in the table below (Table 5).

### 4.6. Limitations and Future Studies

We had some methodological restrictions, and it was not possible to obtain sufficient plasma to measure insulin, and, therefore, the effect of the ingredients on insulin resistance could not be evaluated. Moreover, it was not possible to evaluate browning in the group of CFPulp-E, since we could not obtain sufficient tissue for Mitrotracker staining. Thirdly, we could not assess the amount of food ingested by the mice, and, thus, a possible effect on satiety of the tested ingredients, due to unforeseen methodological problems. Regarding future directions, we think it would be interesting to evaluate if coffee pulp or cocoa shell ingredients are useful for improving the cardiovascular alterations associated with obesity. We have previous evidence that HFD in mice does not induce hypertension but produces endothelial dysfunction and vascular remodeling through an oxidative stress process [33,69]. Our previous results also evidence that cocoa shell extract and some of the bioactive components present in the food matrices tested (protocatechuic acid and caffeine) improved vasodilatation in arteries from aged animals, through an antioxidant action [11]. These data support additional cardiovascular benefits of the ingredients, which deserve further study. Another unexplored and interesting area, a subject of future study, is a metabolomic analysis evaluating the final products generated by the ingredients and their contribution to the observed biological effects. Our previous data with the food matrices tested have demonstrated the colonic biotransformation of non-absorbed phenolic compounds, which could generate smaller and potentially bioactive different metabolites in both coffee pulp [10] and cocoa shell [70]. Therefore, it would be of interest to evaluate the metabolites generated by the ingestion of these by-products.

## 5. Conclusions

The present study demonstrates that coffee pulp exhibits the best efficacy against metabolic syndrome alterations induced by a high-fat diet in female mice and the capacity to normalize body fat and glucose homeostasis and to modify plasma lipids towards a better profile. We suggest that the benefits of coffee pulp ingredients are due to the higher polyphenol content, particularly chlorogenic and protocatechuic acids, which, together with caffeine, may be responsible for the observed bioactivities. The demonstration that these bioactive components are present after simulated digestion supports their capacity to reach plasma. This study provides support to conduct a clinical trial with coffee pulp, since this by-product has recently been accepted as an ingredient for preparing non-alcoholic drinks and infusions pursuant to an E.U. regulation. Therefore, coffee pulp may be used to generate supplements and foods aimed at the control of obesity-related alterations, also contributing to the circular economy.

## Figures and Tables

**Figure 1 foods-12-02708-f001:**
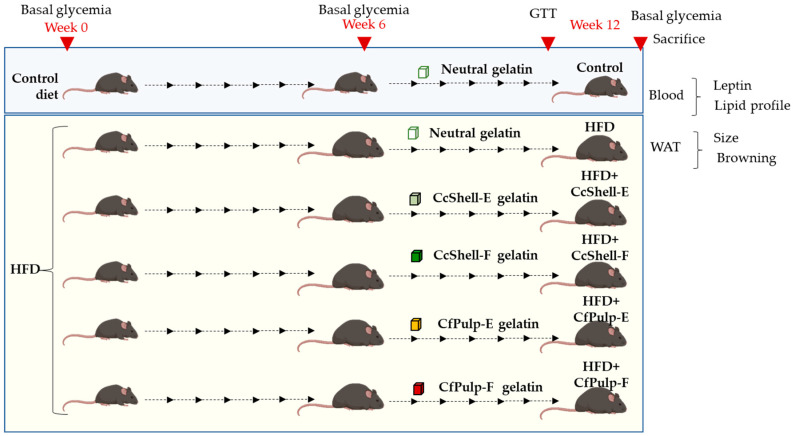
Graphical summary of the experimental design. HFD, high-fat diet; GTT, glucose tolerance test; CfPulp-E, coffee pulp extract; CfPulp-F, coffee pulp flour; CcShell-E, cocoa shell extract; CcShell-F, cocoa shell flour; WAT, white adipose tissue. Arrows represent weeks.

**Figure 2 foods-12-02708-f002:**
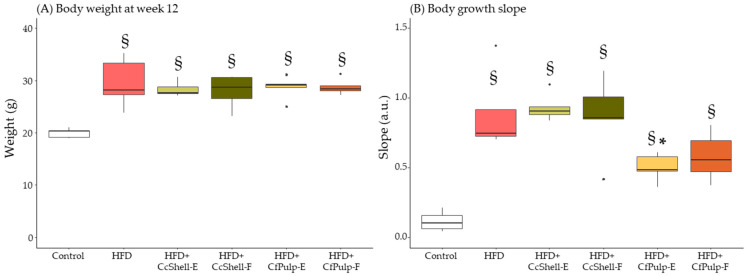
Effect of cocoa shell and coffee pulp ingredients on body weight (**A**), and slope of body growth (**B**) in mice fed HFD. Data show the median with interquartile range [Q1; Q3]. Statistical analysis performed by Kruskal–Wallis and Hold post hoc test. * *p*-value < 0.05 with respect to HFD, ^§^
*p*-value < 0.05 with respect to Control; *n* = 5 mice/group. HFD, high-fat diet; CcShell-E, cocoa shell extract; CcShell-F, cocoa shell flour; CfPulp-E, coffee pulp extract; CfPulp-F, coffee pulp flour.

**Figure 3 foods-12-02708-f003:**
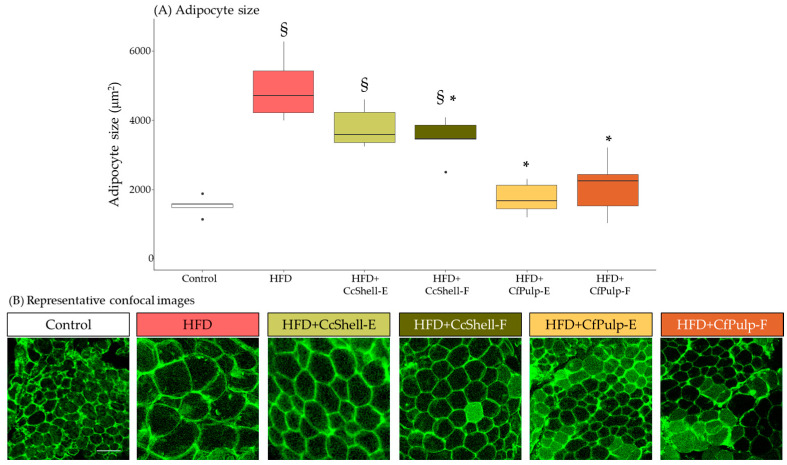
Effect of cocoa shell and coffee pulp ingredients on adipocyte size in mice fed HFD (**A**), and representative confocal images (**B**). Data show the median with interquartile range [Q1; Q3]. Statistical analysis performed by Kruskal–Wallis and Hold post hoc test. * *p*-value < 0.05 with respect to HFD, ^§^
*p*-value < 0.05 with respect to Control; *n* = 5 mice/group. Images were taken with and spectral confocal microscope at Ex. 488 nm/Em. 500–560 nm wavelength with a ×40 objective; image bar, 75 μm. HFD, high-fat diet; CcShell-E, cocoa shell extract; CcShell-F, cocoa shell flour; CfPulp-E, coffee pulp extract; CfPulp-F, coffee pulp flour.

**Figure 4 foods-12-02708-f004:**
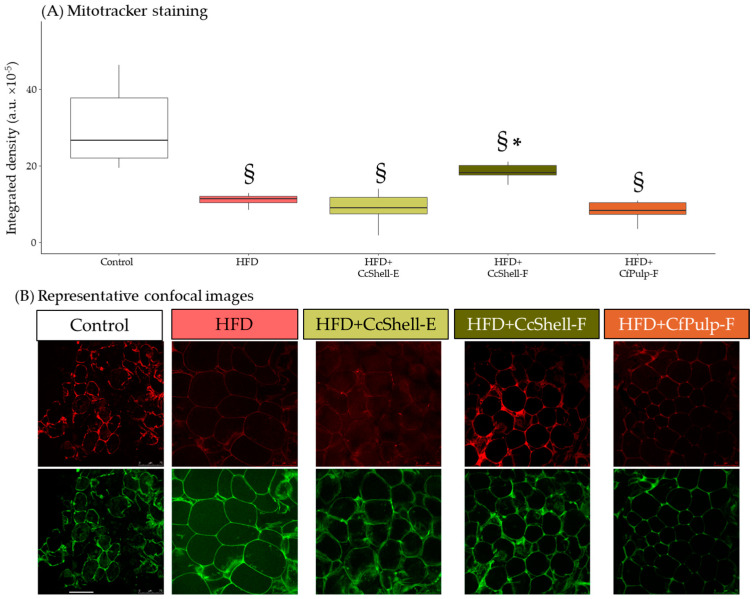
Effect of cocoa shell and coffee pulp ingredients (except CfPulp-E) on Mitotracker staining in adipocytes (**A**) and representative confocal images (**B**) in mice fed HFD. Data show the median with interquartile range [Q1; Q3]. Statistical analysis performed by Kruskal-Wallis and Hold post hoc test. * *p*-value < 0.05 with respect to HFD, ^§^
*p*-value < 0.05 with respect to Control; *n* = 5 mice/group. CcShell-E, cocoa shell extract; CcShell-F, cocoa shell flour; CfPulp-E, coffee pulp extract; CfPulp-F, coffee pulp flour. Images were taken with and spectral confocal microscope at Ex. 488 nm/Em. 500–560 (autofluorescence) and at Ex. 581 nm/Em. 644 nm (Mitotracker); image bar, 75 μm.

**Figure 5 foods-12-02708-f005:**
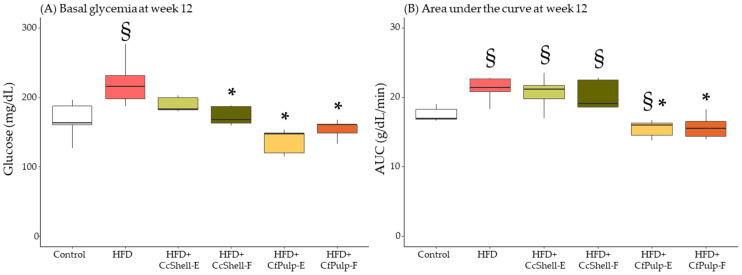
Effect of cocoa shell and coffee pulp ingredients on basal glycemia (**A**), and area under the curve after glucose tolerance test (**B**). Data show the median with interquartile range [Q1; Q3]. Statistical analysis performed by Kruskal–Wallis and Hold post hoc test. * *p*-value < 0.05 with respect to HFD, ^§^
*p*-value > 0.05 with respect to Control; *n* = 5 mice/group. AUC, area under the curve; GTT, glucose tolerance test; CcShell-E, cocoa shell extract; CcShell-F, cocoa shell flour; CfPulp-E, coffee pulp extract; CfPulp-F, coffee pulp flour.

**Figure 6 foods-12-02708-f006:**
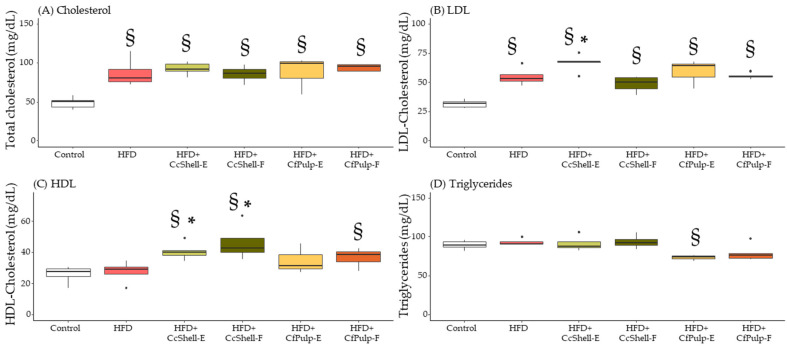
Effect of cocoa shell and coffee pulp ingredients on plasma levels of total cholesterol (**A**), HDL (**B**), LDL (**C**), and triglycerides (**D**). Data show the median with interquartile range [Q1; Q3]. Statistical analysis performed by Kruskal–Wallis and Hold post hoc test. * *p*-value < 0.05 with respect to HFD, ^§^
*p*-value < 0.05 with respect to Control; *n* = 5 mice/group; LDL, low-density lipoprotein-cholesterol; HDL, high-density lipoprotein cholesterol; CcShell-E, cocoa shell extract; CcShell-F, cocoa shell flour; CfPulp-E, coffee pulp extract; CfPulp-F, coffee pulp flour.

**Table 1 foods-12-02708-t001:** Concentration of individual phenolic compounds and methylxanthines in raw material and after in vitro gastrointestinal digestion.

Compound	Raw (mg/100 g)				Gastrointestinal Digested Fraction			
	Cf Pulp-E	Cf Pulp-F	Cc Shell-E	Cc Shell-F	Cf Pulp-E	Cf Pulp-F	Cc Shell-E	Cc Shell-F
**Hydroxybenzoic acids**								
Gallic acid	68.5 ± 4	47.0 ± 2.0	73.9 ± 2.2	16.0 ± 0.4	22.3 ± 2.3 *	14.4 ± 2.0 *	39.2 ± 2.6 *	11.8 ± 0.7 *
Protocatechuic acid	312.2 ± 7.5	175.7 ± 0.7	34.9 ± 2.4	6.5 ± 0.6	221.3 ± 25.8 *	161.6 ± 24.0	33.4 ± 2.3	4.6 ± 0.2 *
**Hydroxycinnamic acids**								
Chlorogenic acid	12.1 ± 1.0	5.3 ± 0.6	-	-	10.5 ± 1.4	6.5 ± 0.4	-	-
Cryptochlorogenic acid (*cis*)	12.7 ± 0.6	6.6 ± 0.5	-	-	14.5 ± 0.4	8.1 ± 1.2	-	-
Cryptochlorogenic acid (*trans*)	145.0 ± 4.0	85.3 ± 1.4	-	-	65.2 ± 0.8 *	67.7 ± 7.3 *	-	-
Neochlorogenic acid	-	3.9 ± 0.3	-	-	-	2.8 ± 0.4	-	-
Caffeic acid	-	-	-	-	-	-	-	-
5-Feruloylquinic acid	8.7 ± 0.1	2.0 ± 0.1	-	-	-	-	-	-
*p*-Coumaric acid	-	-	-	-	-	-	-	-
Isochlorogenic acid A	-	4.9 ± 0.1	-	-	6.9 ± 0.8	3.9 ± 0.4	-	-
3,5-Dicaffeoylquinic acid	7.0 ± 0.1	-	-	-	t	-	-	-
5-*p*-Coumaroylquinic acid	t	2.8 ± 0.1	-	-	-	2.3 ± 0.1	-	-
N-Caffeoyl-L-aspartate	-	-	19.1 ± 1.8	5.1 ± 0.2	-	-	t	t
**Flavones**								
Vicenin-2	8.0 ± 1.1	6.4 ± 0.2	2.9 ± 0.1	-	-	2.7 ± 0.4 *	1.9 ± 0.2 *	-
**Flavan-3-ols**								
Catechin	-	-	46.0 ± 0.1	11.5 ± 0.8	-	-	40.3 ± 0.2 *	16.4 ± 1.3 *
Epicatechin	-	-	3.5 ± 0.1	1.3 ± 0.1	-	-	3.3 ± 0.3	–
**Flavonols**								
Rutin	14.5 ± 0.2	7.9 ± 0.2	-	-	-	4.6 ± 0.1 *	-	-
Isoquercetin	8.9 ± 0.2	6.1 ± 0.3	-	-	t	3.5 ± 0.5 *	-	-
Quercetin 3-*O*-hexoside	-	3.0 ± 0.1	1.4 ± 0.1	0.3 ± 0.0	-	1.6 ± 0.1 *	-	-
Quercetin 3-*O*-pentoside	-	-	1.3 ± 0.0	0.3 ± 0.0	-	-	-	-
* **Total Phenolic Compounds** *	**597.6**	**356.9**	**183.0**	**41.0**	**340.7**	**279.4**	**118.1**	**32.8**
**Methylxantines**								
Caffeine	787.9 ± 34.3	473 ± 6.1	34.0 ± 1.9	169.3 ± 0.3	668.9 ± 70.1 *	388.7 ± 22.1 *	19.9 ± 1.6 *	132.9 ± 3.1 *
Theobromine	-	-	2605.3 ± 125.5	525.8 ± 4.9	-	-	1249.8 ± 49.0 *	510.5 ± 3.1 *

Individual phenolic compounds and methylxanthines (mg/100 g) and total phenolic compounds (mg GAE/g) in raw material and after in vitro gastrointestinal digestion of coffee pulp and cocoa shell ingredients characterized by UPLC–ESI–MS/MS. Data show the mean ± standard deviation. * *p*-value < 0.05 compared to raw material by Tukey’s test. CfPulp-E, coffee pulp extract; CfPulp-F, coffee pulp flour; CcShell-E, cocoa shell extract; CcShell-F, cocoa shell flour; t, traces.

**Table 2 foods-12-02708-t002:** Body weigh at week 1 and at week 6 in Control and HFD-fed groups.

	Control (*n* = 5)	HFD (*n* = 5)	HFD+ CcShell-E (*n* = 5)	HFD+ CcShell-F (*n* = 5)	HFD+ CfPulp-E (*n* = 5)	HFD+ CfPulp-F (*n* = 5)
Weight at week 1 (g)	19.9 [18.9; 20.1]	19.5 [19.3; 19.8]	18.9 [18.6; 18.9]	19.3 [18.7; 19.5]	21.8 [21.2; 21.8]	20.7 [20.2; 21.1]
Weight at week 6 (g)	20.3 [19.2; 20.3]	25.3 [24.6; 29.3] ^§^	25.0 [24.8; 25.3] ^§^	25.3 [22.7; 28.1] ^§^	24.4 [24.2; 26.4] ^§^	25.3 [22.0; 25.6] ^§^

Data show median and interquartile range [Q1; Q3]. Statistical analysis by Kruskal–Wallis and Hold post hoc test; ^§^
*p*-value < 0.05 when compared to Control; n, number of mice. HFD, high-fat diet; CcShell-E, cocoa shell extract; CcShell-F, cocoa shell flour; CfPulp-E, coffee pulp extract; CfPulp-F, coffee pulp flour groups before supplementation.

**Table 3 foods-12-02708-t003:** Leptin levels in Control and HFD-fed mice after supplementation.

Group	Control (*n* = 4)	HFD (*n* = 4)	HFD+ CcShell-E (*n* = 4)	HFD+ CcShell-F (*n* = 4)	HFD+ CfPulp-E (*n* = 4)	HFD+ CfPulp-F (*n* = 4)
Leptin (pg/dL)	3.46 [3.41; 3.55]	45.5 [40.6; 50.1] ^§^	30.1 [25.9; 31.8] ^§^	31.5 [29.0; 36.5] ^§^	21.5 [19.5; 23.6] ^§^	31.0 [25.9; 31.8] ^§^

Data show median and interquartile range [Q1; Q3]. Statistical analysis by Kruskal–Wallis and Hold post hoc test; ^§^
*p*-value < 0.05 when compared to Control; *n*, number of mice. CcShell-E, cocoa shell extract; CcShell-F, cocoa shell flour; CfPulp-E, coffee pulp extract; CfPulp-F, coffee pulp flour.

**Table 4 foods-12-02708-t004:** Basal glycemia at week 1 and at week 6 in Control and HFD-fed groups.

Group	Control (*n* = 5)	HFD (*n* = 5)	HFD+ CcShell-E (*n* = 5)	HFD+ CcShell-F (*n* = 5)	HFD+ CfPulp-E (*n* = 5)	HFD+ CfPulp-F (*n* = 5)
Week 1 (mg/dL)	171 [162; 186]	173 [160; 176]	173 [160; 176]	184 [183; 197]	161 [161; 184]	153 [143; 169]
Week 6 (mg/dL)	163 [163; 168]	216 [198; 231] ^§^	200 [190; 203] ^§^	184 [183; 197] ^§^	201 [201; 209] ^§^	216 [190; 222] ^§^

Data show median and interquartile range [Q1; Q3]. Statistical analysis by Kruskal–Wallis and Hold post hoc test; ^§^
*p*-value < 0.05 when compared to Control; *n*, number of mice. CcShell-E, cocoa shell extract; CcShell-F, cocoa shell flour; CfPulp-E, coffee pulp extract; CfPulp-F, coffee pulp flour groups before supplementation.

**Table 5 foods-12-02708-t005:** Summary of the effects of the tested cocoa shell and coffee pulp ingredients to improve the metabolic alterations induced by HFD.

**Parameter**	**Cocoa Shell**		**Coffee Pulp**	
	Extract	Flour	Extract	Flour
Body growth rate			+	
Adipocyte size		+	++	++
Adipocyte browning		+		
Leptin			+	
Basal glycemia		++	++	++
Glucose tolerance			++	++
Cholesterol				
LDL				
HDL	+	+		+
Triglycerides			+	

HFD, high-fat diet; CfPulp-E, coffee pulp extract; CfPulp-F, coffee pulp flour; CcShell-E, cocoa shell extract; CcShell-F, cocoa shell flour. “+” indicates an improvement and “++” normalization of the parameter.

## Data Availability

The data presented in this study are available upon request from the corresponding author. The availability of the data is restricted to investigators based in academic institutions.

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
