# Peer review of "Effect of Supplementation with Coffee and Cocoa By-Products to Ameliorate Metabolic Syndrome Alterations Induced by High-Fat Diet in Female Mice"

_foods, 2023, doi:10.3390/foods12142708_

Round 1

Reviewer 1 Report

The overall design of the paper is relatively reasonable, and the good improvement effect of coffee production by-products on metabolic syndrome was studied. However, there are still some details that need to be noted.

Line 101: This section should be split into two parts. Preparation, Characterization

Line116: Please unify the horizontal bars.

Line 125: What is the significance of in vitro digestion experiments? Can you delete it? After deletion, it is actually more beneficial for readers to read

Line 129: Please confirm if the symbol for Celsius is correct.

Line 143: Is the number of five mice in each group too small?

Line 179: The accuracy of title expression needs to be improved

Line 201: Please unify all negative signs throughout the entire text

Line 287: Two decimal places should be retained; Total Phenolic Compounds-There should be a standard deviation.

Line 832: Remove horizontal bars from names

Line 865: Remove the E mark from the page number

Line 869: Where is the title of the article

The writing of the paper is relatively standard, and it is suggested to convert the Active voice into the Passive voice

Author Response

The overall design of the paper is relatively reasonable, and the good improvement effect of coffee production by-products on metabolic syndrome was studied. However, there are still some details that need to be noted.

  • Line 101: This section should be split into two parts. Preparation, Characterization

ANSWER: Thank you for your comments. Regarding this point, we have separated the section 2.1. from Material and Methods, as suggested. We have also separated Materials from methods, as suggested by another reviewer.

  • Line116: Please unify the horizontal bars.

ANSWER: We have modified the horizontal bars and homogenized throughout the text. The only exception is the bars separating the numbers of the references within the text, since we cannot modify this.

  • Line 125: What is the significance of in vitro digestion experiments? Can you delete it? After deletion, it is actually more beneficial for readers to read.

ANSWER: The simulated digestion of the matrices used in the study mimic gastrointestinal digestion and, thus, represent the compounds which are more likely to reach plasma and can be the contributors to the observed changes. Therefore, we think that the data on simulated digestion is relevant, and we would prefer maintaining this part, also in accordance with other reviewers comments.

  • Line 129: Please confirm if the symbol for Celsius is correct.

ANSWER: Yes, it is correct.

  • Line 143: Is the number of five mice in each group too small?

ANSWER: We are following the protocols of 3Rs of our institution Ethical Committee, regarding reduction of experimental animals whenever possible. Since we have detected significant differences with five mice, we do not think an increase is needed.  

  • Line 179: The accuracy of title expression needs to be improved.

ANSWER: We have changed it to: “Graphical summary of the experimental design”

  • Line 201: Please unify all negative signs throughout the entire text.

ANSWER: As indicated, we have unified the negative signs (horizontal bars) with the exception of the numbers of references within the text.

  • Line 287: Two decimal places should be retained; Total Phenolic Compounds-There should be a standard deviation.

ANSWER: We do not think this is relevant and it would be difficult to fit 2 decimals in the table. On the other hand, in Table 1 we have shown the results as mean± standard deviation, and highlighted it in the legend, as suggested.

  • Line 832: Remove horizontal bars from names.

ANSWER: We have removed the horizontal bars

  • Line 865: Remove the E mark from the page number.

ANSWER: We have removed the E mark

  • Line 869: Where is the title of the article.

ANSWER: The title of the article is “Coffee, Hunger and Peptide YY”.

ANSWER: We have corrected the entire style

Reviewer 2 Report

After careful review, there are some problems in the design of this manuscript, and some major concerns. In my opinion, the manuscript needs to be modified completely, otherwise it is unavoidable to be rejected.

1. there were little results about “Ameliorate Metabolic Syndrome”. Some metabonomic results must be added.

2. the bioactive composition of Coffee Pulp and Cocoa Shell extracts and flours must have more composition than in Table 1. So, why?

3. CfPulp-E, CfPulp-F, CcShell-E and CcShell-F are not medicine. So, if supplementation protocol from first week maybe a better result?

Moderate editing of English language required.

Author Response

After careful review, there are some problems in the design of this manuscript, and some major concerns. In my opinion, the manuscript needs to be modified completely, otherwise it is unavoidable to be rejected.

  1. there were little results about “Ameliorate Metabolic Syndrome”. Some metabonomic results must be added.

ANSWER: Thank you for your comments. We think we have explored the main features of metabolic syndrome in the study, including adipocyte size, browning, leptin, glucose homeostasis (basal glycemia and glucose tolerance) and lipid profile.

We agree on the importance of evaluating the metabolomic profile to assess major modifications induced by the tested food matrices. In fact, previous simulated digestion studies evidence the potential colonic transformation of both Cocoa Shell (doi: 10.1016/j.foodres.2022.112117) and Coffee Pulp (doi: 10.3390/antiox11091818), which fully supports your suggestion. However, the present study was designed as a first step to evaluate which of the food ingredients is the best option for applications in the context of Metabolic Syndrome. Based on our results we propose that Coffee Pulp is the best by-product, and therefore, we would like to continue with our studies with this ingredient performing a clinical trial, where the assessment of metabolomic study is very pertinent. We fully understand the importance of metabolomics and, accordingly, we have included this aspect in the Limitations of the study and Future Studies section.

  1. the bioactive composition of Coffee Pulp and Cocoa Shell extracts and flours must have more composition than in Table 1. So, why?

ANSWER: We have previously fully characterized these food matrices. In the present study we have included the phenolic profile in raw and intestinal digested fraction which are associated with the improvement of the metabolic syndrome alterations induced, with the aim of providing a comprehensive explanation of the observed effects. In our previous studies with Coffee Pulp a total of 17 phenolic compounds were identified, as well as caffeine (doi: 10.3390/antiox11091818 ). Regarding Cocoa Shell, we have previously optimized the extraction and fully characterized the bioactive components of this by-product, showing a wide range of Hydroxybenzoic and hydroxycinnamic acids, mandelic and phenylacetic acids, as well as flavan-3-ols, and flavanols (doi: 10.1016/j.seppur.2021.118779). We have included some additional information in the new version.

CfPulp-E, CfPulp-F, CcShell-E and CcShell-F are not medicine. So, if supplementation protocol from first week maybe a better result?

ANSWER: Yes, we agree. However, the objective of the study was to evaluate the capacity of these by-products to reduce the alterations caused by high fat diet, once the metabolic syndrome alterations are already established, rather than to evaluate the capacity to prevent them. We decided to choose this objective, since in humans high-fat diet induced MetS is very prevalent. We think these food ingredients can be used in conjunction with dietary, or pharmacological interventions, to reduce the impact of obesity.  We agree that it would also be interesting to test the capacity of Coffee Pulp to prevent fat accumulation. We thank you for this suggestion, which can be carried out in future studies in mice supplemented together with HFD.

ANSWER: We have revised the text and amended grammatical errors and typos. 

Reviewer 3 Report

This study discusses the application of Coffee pulp or Cocoa shell extract/flour on improving the symptoms of metabolic syndrome in female mice. The topic of the manuscript is interesting and suitable for the journal’s readership. However, the manuscript still should be improved in the Methodology section, and additional information should be provided relevant to the followed procedures and protocols. In addition, in some sections of the manuscripts, sentences do not read well and should be restructured.

I have the following comments for the authors:

1.    In the section “2. Materials and Methods”, a subsection should be added for "Materials" separately and the authors should list all the materials and chemicals used and their suppliers including the enzymes used for the in vitro simulated digestion.

The Materials should be separated from the preparation methods.

2.     Additional information is required to be provided in the methodology section:

·      Sentences in lines 112-114, require to be restructured. The chromatograph does not include a pump! The sentence does not read well as is currently presented.

Do authors intend to mention HPLC systems (Agilent technology) with a quaternary pump and DAD?

·      In lines 136-137, section 2.3,  Explicitly state if there were any inclusion or exclusion criteria.

·      In lines 157-158, explain the procedure followed to reduce pain or relief stress before sacrificing mice. Refer to a standard protocol and add details in the text.

3.     In line 38 of the abstract section, provide the complete term for “HDL”. Similarly in line 201 complete terms for “HDL” and “LDL” should be provided the first time mentioned in the text and an abbreviation should be used afterwards. Same for “DAPI” in line 225.

4.     In the keywords section, remove the punctuation mark “.” After the keywords. In line 276, a punctuation mark is missing in line 276 after “degree of persistence”.

5.    In lines 49 and, 501 “Pandemic obesity” is used incorrectly. The word "Pandemic" is a widespread occurrence of an infectious disease over a whole country or worldwide.

Obesity is not an infectious disease. Therefore, "pandemic of obesity" is used incorrectly by the authors. This section should be modified and rewritten.

6.     Minor grammatical mistakes should be corrected:

·      In line 52, “linked with oxidative stress” should change to “linked to oxidative stress”.

·      In line 56, “have been subject” should change to “have been the subject”

·      Inline 240, “significant” should change to “significance”

7.     Some sentences require to be restructured and do not read well:

·      In line 242, the sentence:” For all the analysis was considered significant a p-value<0.05.”

This sentence requires to be restructured. It is grammatically inaccurate.

·      Line 520, the sentence: “since the solubilization of phytochemicals from the ingredients was produced by” requires to be restructured. "Solubilization" is not produced by heat extraction. Heat-assisted extraction facilitates solubilization.

8.     Some errors and mistakes in the presented tables should be corrected:

·      In Table 1, “Hydroxybendoic acid” should change to “Hydroxybenzoic acid”

·      In Table 4, CfPulp-F is duplicated. One should be distinguished as CfPulp-E

9.     In line 465, Did the authors intend to refer to Figure 6B instead of Figure 6C? Similarly, in line 467, it seems that Figure 6C should be referred to instead of Figure 6B. Please check and correct.

Some sentences require to be restructured as they are presented inaccurately and do not read well. Some other minor grammatical mistakes should also be corrected. I have enclosed my comments to the authors.

Author Response

This study discusses the application of Coffee pulp or Cocoa shell extract/flour on improving the symptoms of metabolic syndrome in female mice. The topic of the manuscript is interesting and suitable for the journal’s readership. However, the manuscript still should be improved in the Methodology section, and additional information should be provided relevant to the followed procedures and protocols. In addition, in some sections of the manuscripts, sentences do not read well and should be restructured.

 I have the following comments for the authors:

  1. In the section “2. Materials and Methods”, a subsection should be added for "Materials" separately and the authors should list all the materials and chemicals used and their suppliers including the enzymes used for the in vitro simulated digestion. The Materials should be separated from the preparation methods.

ANSWER: Thank you for your comments. We have now amended Materials and Methods as suggested.

  1. Additional information is required to be provided in the methodology section:
    1. Sentences in lines 112-114, require to be restructured. The chromatograph does not include a pump! The sentence does not read well as is currently presented.
    2. Do authors intend to mention HPLC systems (Agilent technology) with a quaternary pump and DAD?

ANSWER: We have corrected the methodological section according to the suggestions.

  1. In lines 136-137, section 2.3, explicitly state if there were any inclusion or exclusion criteria.

ANSWER: Our inclusion criteria was a similar initial body weight and age. We have made this explicit in the suggested text (section “Experimental design in mice”).

  1. In lines 157-158, explain the procedure followed to reduce pain or relief stress before sacrificing mice. Refer to a standard protocol and add details in the text.

ANSWER: In our study we used CO2 –induced hypoxia as the standard method used to avoid pain during cardiac puncture, approved by our Ethical Committee. We have made this explicit in the text.

  1. In line 38 of the abstract section, provide the complete term for “HDL”. Similarly in line 201 complete terms for “HDL” and “LDL” should be provided the first time mentioned in the text and an abbreviation should be used afterwards. Same for “DAPI” in line 225.

ANSWER: We have amended both.

  1. In the keywords section, remove the punctuation mark “.” After the keywords. In line 276, a punctuation mark is missing in line 276 after “degree of persistence”.

ANSWER: We have amended these typos.

  1. In lines 49 and, 501 “Pandemic obesity” is used incorrectly. The word "Pandemic" is a widespread occurrence of an infectious disease over a whole country or worldwide. Obesity is not an infectious disease. Therefore, "pandemic of obesity" is used incorrectly by the authors. This section should be modified and rewritten.

ANSWER: We think pandemic is a correct word, but we probably we used it in an incorrect way. If used as noun means “a widespread occurrence of an infectious disease over a whole country or the world at a particular time”. However, it can be used as adjective (of a disease, in this case obesity), meaning “prevalent over a whole country or the world”. This exception is widely used in the literature in the context of obesity, including by WHO (doi: 10.1016/S2468-1253(21)00143-6) . We have corrected the sentence to “The prevalence of sedentary lifestyle and inadequate diets has contributed to the development of obesity, which is currently considered a pandemic disease” and also eliminated this word from the discussion.

  1. Minor grammatical mistakes should be corrected:
    1. In line 52, “linked with oxidative stress” should change to “linked to oxidative stress”.
    2. In line 56, “have been subject” should change to “have been the subject”
    3. In line 240, “significant” should change to “significance”
  2. Some sentences require to be restructured and do not read well:
    1. In line 242, the sentence: “For all the analysis was considered significant a p-value<0.05.” This sentence requires to be restructured. It is grammatically inaccurate.
    2. Line 520, the sentence: “since the solubilization of phytochemicals from the ingredients was produced by” requires to be restructured. "Solubilization" is not produced by heat extraction. Heat-assisted extraction facilitates solubilization.
  3. Some errors and mistakes in the presented tables should be corrected:
    1. In Table 1, “Hydroxybendoic acid” should change to “Hydroxybenzoic acid”
    2. In Table 4, CfPulp-F is duplicated. One should be distinguished as CfPulp-E
  4. In line 465, Did the authors intend to refer to Figure 6B instead of Figure 6C? Similarly, in line 467, it seems that Figure 6C should be referred to instead of Figure 6B. Please check and correct.

ANSWER: Thank you for the detection of grammatical errors and typos. We have corrected them in the new version and revised the text to improve the English Language.

Reviewer 4 Report

The authors present a very interesting and timely study investigating the effects of supplementation with coffee and cocoa by-products on metabolic syndrome alterations induced by a high-fat diet in female mice. The article is well-written, the methodology is clearly outlined, and the conclusions drawn align with the results presented.

However, there are a few minor issues that could be addressed further to enhance the clarity and impact of this article:

1. Uniformity in Terminology: The terms 'Coffee Pulp' and 'Cocoa Shell' are inconsistently capitalized throughout the article. It is suggested that the authors adopt a uniform style to enhance readability and consistency.

2. Clarity in the Materials and Methods Section: In the Materials and Methods section, lines 111 to 124 provide information regarding the injection volume and standards used for analysis.

3. Clarity on the Sex of the Mice: As the article focuses on female mice, the discussion section must indicate the sex of the mice used in each experiment for comparison. This will help assess the relevance and applicability of the findings about gender differences in metabolic syndrome.

4. Inclusion of Hypothetical Outcomes in Male Mice: The authors could discuss whether they expect to observe different results if male mice were used in the study. Including hypothetical outcomes and a rationale for potential differences between sexes would enrich the discussion. For instance, hormonal differences between male and female mice may influence the metabolic response to diet and supplementation.

5. Specificity of Food Supplements for Females: The authors might explore whether the food supplements would have specific benefits for females due to, for instance, hormonal differences that affect metabolism. It is important to discuss whether these findings can be generalized or if they are specifically applicable to female mice.

6. Implication of Findings for Both Sexes: Lastly, discussing what might change if both sexes use these supplements is crucial. The authors should discuss how incorporating male mice in future studies could strengthen or alter the current findings. Moreover, understanding how these supplements affect both sexes can have important implications for developing interventions for metabolic syndrome in humans.

The quality of English language usage is good, with only minor editing necessary to correct a few typographical errors.

Author Response

The authors present a very interesting and timely study investigating the effects of supplementation with coffee and cocoa by-products on metabolic syndrome alterations induced by a high-fat diet in female mice. The article is well-written, the methodology is clearly outlined, and the conclusions drawn align with the results presented. However, there are a few minor issues that could be addressed further to enhance the clarity and impact of this article:

  1. Uniformity in Terminology: The terms 'Coffee Pulp' and 'Cocoa Shell' are inconsistently capitalized throughout the article. It is suggested that the authors adopt a uniform style to enhance readability and consistency.

ANSWER: Thank you for your comments. We have made this terms uniform through the text.

  1. Clarity in the Materials and Methods Section: In the Materials and Methods section, lines 111 to 124 provide information regarding the injection volume and standards used for analysis.

ANSWER: We have included this information in the text

  1. Clarity on the Sex of the Mice: As the article focuses on female mice, the discussion section must indicate the sex of the mice used in each experiment for comparison. This will help assess the relevance and applicability of the findings about gender differences in metabolic syndrome.

ANSWER: We have now included the sex of the experimental animals in other studies, to compare with our study.  

  1. Inclusion of Hypothetical Outcomes in Male Mice: The authors could discuss whether they expect to observe different results if male mice were used in the study. Including hypothetical outcomes and a rationale for potential differences between sexes would enrich the discussion. For instance, hormonal differences between male and female mice may influence the metabolic response to diet and supplementation.

ANSWER: We have added a paragraph in the discussion regarding the potential effects of the by-products if male mice were used, comparing our results with those in male experimental animals using Coffee Pulp extracts and evaluating the influence of sex hormones (lines 711-716).

  1. Specificity of Food Supplements for Females: The authors might explore whether the food supplements would have specific benefits for females due to, for instance, hormonal differences that affect metabolism. It is important to discuss whether these findings can be generalized or if they are specifically applicable to female mice.
  2. Implication of Findings for Both Sexes: Lastly, discussing what might change if both sexes use these supplements is crucial. The authors should discuss how incorporating male mice in future studies could strengthen or alter the current findings. Moreover, understanding how these supplements affect both sexes can have important implications for developing interventions for metabolic syndrome in humans.

ANSWER: Point 5 and 6 are very important questions, particularly in future studies in humans. We have discussed differences in MetS alterations between men and women and the potential benefits of the tested by-products in both sexes, based on these differences since our next step would be to develop clinical trials with Coffee Pulp ingredients.

ANSWER: We have revised the text and corrected errors.

Reviewer 5 Report

Dear Authors,

You are working on a trending topic of research. However, you can introduce some changes in your manuscript:

- please, revise format of manuscript according to journals guidelines;

- revise conclusions section to make it more informative

Please, double-check for minor mistakes.

Author Response

Dear Authors, you are working on a trending topic of research. However, you can introduce some changes in your manuscript:

  • please, revise format of manuscript according to journals guidelines.
  • revise conclusions section to make it more informative.

ANSWER: Thank you for your comments. We have modified the manuscript according to the guidelines and also modified the conclusion section to make it more informative.

ANSWER: We have revised the text and amended errors. 

Reviewer 6 Report

This manuscript aims to conduct a comparative study to evaluate the efficacy of supplementation with flours and extracts derived from Cocoa Shell and Coffee Pulp to reduce MetS alterations induced by HFD, to determine which is the ingredient with best potential for future human applications. In HFD-fed female mice the capacity of these four ingredients to reduce body weight, adiposity, hyperglycemia, and hyperlipidemia were compared.

The study provides information regarding the effectiveness of coffee pulp as potential ingredient to  ameliorate metabolic syndrome alterations induced by high fat diet and suggests that it can be incorporated by the food industry to generate supplements and foods, which can help to control obesity- related metabolic alteration, also contributing to the circular economy.

Minor editing of English language required

Author Response

ANSWER: Thank you for your comments. We have revised the text and amended errors.

Round 2

Reviewer 2 Report

it can be  accepted

it can be  accepted